# Community health worker intervention to reduce worker exposure to volatile organic compounds in small business auto and beauty shops in a marginalized community: A cluster randomized controlled trial

Shannon L. Gutenkunst[1]*, Nathan Lothrop[2], Carolina Quijada[2], Marvin Chaires[2], Imelda Cortez[3], Flor Sandoval[3], Jella Balgos[3], Emma V. Gallardo[2], Pedro Flores Gallardo[2], Sam Sneed[2], Jenna K. Honan[2], Karla Bayless[2], Xavier Chaidez[2], Cristobal Reyes Cuevas[2], Jacqueline L. Larson[2], Fernanda J. Camargo[2], Denise Moreno Ramírez[2,4], Maia Ingram[2], Scott Carvajal[2], Dean Billheimer[1,2], Ann Marie Wolf[3], Paloma I. Beamer[2,5]*

1 Center for Biomedical Informatics and Biostatistics, Statistics Consulting Laboratory, The University of Arizona, Tucson, Arizona, United States of America, 2 Mel and Enid Zuckerman College of Public Health, The University of Arizona, Tucson, Arizona, United States of America, 3 Sonora Environmental Research Institute, Incorporated, Tucson, Arizona, United States of America, 4 Julie Ann Wrigley Global Futures Laboratory, Arizona State University, Tempe, Arizona, United States of America, 5 BIO5 Institute, The University of Arizona, Tucson, Arizona, United States of America

* shannonlg@arizona.edu (SLG); pbeamer@arizona.edu (PIB)

## Abstract

### Background

Occupational diseases affect many workers in the United States, with Latinos disproportionately affected. Small businesses face barriers to implementing workplace health protections that community health workers (CHWs) may help overcome. The objective of this study was to determine whether a CHW-led industrial hygiene intervention could reduce volatile organic compound (VOC) exposure in small auto repair and beauty shops that primarily employ marginalized workers.

### Methods

In this two-arm, parallel, cluster randomized trial, small business (≤25 employees) auto repair and beauty shops in Tucson, AZ were randomized to immediate or delayed intervention, stratified by sector. CHWs assessed shops and provided knowledge of controls and $300 for new ones. Total VOCs (TVOCs) were measured using photoionization detectors placed on or near participants. The primary outcome was the change in TVOCs at the shop level after the intervention, assessed across three timepoints with four worshift measurements per assessment. Mixed-effects models accounted for clustering by shop.

**Data availability statement:** Individual participant data cannot be made publicly available to protect participant privacy. The informed consent process did not include permission for public data sharing, and the dataset contains sensitive workplace and demographic information that could potentially identify participants in small businesses. Data access can be requested using the University of Arizona's Data Use Agreement (DUA) procedures from the Office of Research and Partnerships: https://research.arizona.edu/faq-page/data-use-agreement. Contact email to request a DUA: contracting@email.arizona.edu.

**Funding:** This project was supported by the National Institute of Environmental Health Sciences grants R01 ES028250, P30 ES006694, T32 ES007091, and R25 ES025494. The publication's contents are solely the authors' responsibility and do not necessarily represent the official views of the National Institutes of Health. The funders had no role in study design, data collection and analysis, decision to publish, or preparation of the manuscript. There was no additional external funding received for this study.

**Competing interests:** The authors have declared that no competing interests exist.

**Abbreviations:** ACH, Air changes per hour; AFL-CIO, American Federation of Labor and Congress of Industrial Organizations; ANP, Auto No Paint (shop subtype); ANSI, American National Standards Institute; ASOS, Automated Surface Observing System; ASHRAE, American Society of Heating, Refrigerating, and Air-Conditioning Engineers; BH, Beauty Hair Only (shop subtype); BN, Beauty Hair and Nails (shop subtype); cfm, Cubic feet per minute; CHW, Community health worker; CI, Confidence interval; CONSORT, Consolidated Standards of Reporting Trials; COVID-19, Coronavirus Disease 2019; EPA, Environmental Protection Agency; GM, Geometric mean; GSD, Geometric standard deviation; IPA, Isopropyl alcohol (2-propanol); IQR, Interquartile range; LOD, Limit of detection; NIOSH, National Institute for Occupational Safety and Health; OSHA, Occupational Safety and Health Administration; PID, Photoionization detector; PPE, Personal protective equipment; REDCap, Research Electronic Data Capture; SD, Standard deviation; SERI, Sonora Environmental Research Institute, Inc.; TVOC, Total volatile organic compound; TWA, Time-weighted average; UA, University of Arizona; VOC, Volatile organic compound.

## Results

We enrolled 38 auto repair shops and 46 beauty shops (73% Latino workers) and analyzed 846 workshift measurements at 236 shop assessments. Adjusted models showed a non-statistically significant intervention effect: auto shops experienced on average an estimated 28% TVOC increase (95% CI: −46% to 203%); beauty shops experienced on average an estimated 27% reduction (95% CI: −55% to 19%). Beauty shops had TVOC concentrations about 10 times higher than auto shops, and 87% of their assessments had ventilation rates below the recommended minimum.

## Conclusions

Although not statistically significant, the CHW-led intervention may meaningfully reduce VOC exposure in beauty shops. High TVOC concentrations and inadequate ventilation in beauty shops highlight the need for targeted interventions and policy changes to improve the air quality in these underserved small businesses.

## Trial registration

This trial was registered with clinicaltrials.gov (NCT03455530) on March 6, 2018.

## Introduction

Occupational diseases contribute to approximately 120,000 deaths annually in the United States, accounting for an estimated 3.7% of all deaths [1,2]. Low-income minority workers bear a disproportionate share of this burden [3]; in particular, Latino workers have both a 24% higher rate of workplace deaths than the national average and a 24% increase in workplace deaths over the last decade [1]. Many low-income minority workers are employed by small businesses. The complex interaction of physical, social, and economic factors makes it difficult for small businesses to protect worker health [4]. Compared with larger businesses, small businesses lack access to resources and dedicated personnel for safety and health practices, resulting in higher rates of occupational diseases [5]. The number of small businesses in the US far exceeds the capacity of government programs: each Occupational Safety and Health Administration (OSHA) inspector oversees over 4,000 worksites [6]. Additionally, many small business owners lack the language, literacy, and computer skills to access online safety guidance. Given these challenges, alternative approaches to providing occupational health guidance for marginalized workers in small businesses are needed.

Community health workers (CHWs) offer a promising approach to overcome these challenges, because as trusted members of their communities, CHWs can bridge gaps in access to information by providing culturally appropriate education and support to business owners and workers. CHWs have long been involved in interventions addressing environmental exposure and health. For example, CHW interventions have reduced occupational exposures among farmworkers [7,8], and they

have improved health outcomes in various settings [9,10]. However, their effectiveness in small business settings remains understudied.

In southern Arizona, the Sonora Environmental Research Institute, Inc. (SERI) – a nonprofit community-based organization with over 30 years of experience – mitigates environmental risk through a team of CHWs. Known variously as community outreach workers, community environmental health workers, or in Spanish as *promotoras de salud*, these trusted local liaisons bring expertise in environmental health and safety directly to small business owners and workers across various sectors [11]. Two common small business sectors in the low-income, predominantly Latino neighborhoods in the Tucson, Arizona metropolitan area that SERI has experience working with are auto repair shops and beauty shops.

Understanding occupational health in auto repair and beauty shops is of particular interest for three key reasons: (1) they are prevalent small businesses that use solvents [11]; (2) they allow comparisons between male-dominated and female-dominated workplaces, because gender may influence the use of occupational health guidance [12,13]; and (3) they both commonly use volatile organic compounds (VOCs). Although auto body shops and nail salons are better studied for their known high-VOC activities [14–18], many of the same activities plus others occur in auto repair and beauty shops, and more people work in them: in the US in 2023, there were over four times more auto repair workers than auto body workers, and there were about two times more beauty shop workers than nail salon workers [19].

Despite the health risks associated with VOC exposure in auto repair and beauty shops, few studies have evaluated scalable, shop-level interventions to reduce exposure. The present study addresses this gap by evaluating a CHW-led industrial hygiene intervention tailored to small auto repair and beauty shops. SERI CHWs customized VOC reduction strategies (i.e., controls) to specific shop conditions, engaging owners and workers in selecting easily adoptable controls to improve workplace safety. Controls often included ventilation improvements such as air purifiers or fans, and exposure reduction measures such as trash cans with lids or personal protective equipment (PPE). To test whether such a CHW-led intervention could reduce total VOC exposure (primary outcome) and lower specific VOC hazard scores (secondary outcome) in small auto repair and beauty shops in low-income, predominantly Latino neighborhoods in the Tucson, AZ metropolitan area, we used a cluster randomized trial, in which clusters were shops. This is the first study to use a formal randomized controlled trial to determine whether a CHW-led intervention decreased VOC exposure in small businesses. The success of this scalable intervention model could have broad implications for reducing occupational health disparities across various small business sectors.

## Methods

### Implementation, setting, and participants

A collaboration between SERI, the El Rio Community Health Center (El Rio), and the University of Arizona (UA) made this study possible. Bilingual CHWs from SERI conducted initial participant recruitment and the intervention, and then they followed up with each shop with a report of their VOC exposures after each shop finished the trial. El Rio staff offered free health screenings and provided community resources for follow-up health care and insurance navigation. A bilingual measurement team from UA measured total and specific VOCs and recorded shop and participant characteristics; a separate bilingual UA team conducted final interviews of participants. Statisticians at UA (who had no contact with study participants) provided randomization lists, automated reports of participant exposures and gave them to SERI when shops completed the study, and analyzed the results of the cluster randomized trial after all shops had completed the study.

SERI CHWs conducted recruitment from January 10, 2022 to May 11, 2023. Eligible businesses were small (≤25 employees) auto repair or beauty shops in specific Tucson, AZ ZIP codes with particular demographics and low socio-economic status. Auto repair shops had to offer mechanical repair, with or without autobody repair; those solely providing auto paint/body work were excluded. Beauty shops had to offer hair services, with or without nail or skin care; those solely providing nail or skin care were excluded. Chain/franchised shops and those sharing a single entrance with other shops

were excluded. Each shop received up to $300 to purchase VOC exposure controls of their choice, as informed by CHWs from SERI.

Eligible workers at participating businesses were 18+ years old and able to speak, read, and write Spanish or English. Each participant gave written informed consent after randomization and received $10 cash per workshift for photoionization detector (PID) use, plus their personal total and shop's specific VOC data, free health screenings (blood pressure, glucose, body mass index), and resources for follow-up health care and insurance navigation.

## Ethical approval

Informed consent was obtained from all participants, and study ethics approval was obtained from the University of Arizona's Human Subjects Protection Program (#1709821542), in accordance with the Declaration of Helsinki.

## Trial design and important changes

This was a two-arm, parallel, cluster randomized trial, in which clusters were the shops. The cluster design ensured logistical simplicity and consistency in intervention application across all workers within a shop and minimized contamination risks. Additionally, this design was statistically efficient, because it accounted for within-cluster correlations, which was crucial in these small businesses where VOC exposures are interconnected because of shared workplace environments. Shops were randomized 1:1 to immediate or delayed intervention. To ensure all shops had the same information at baseline, both groups received packets on pollution prevention from a previous project [11] before data collection. Each shop had data collected during three assessments. The immediate intervention group received the CHW-led intervention between the first and second assessments, and the delayed intervention group received it between the second and third assessments (see Fig 1). Both groups were offered health screenings between the first and second assessments and after the final assessment. The trial results have been reported here following the CONSORT statement for cluster randomized trials [20] as closely as possible (see S1 Checklist).

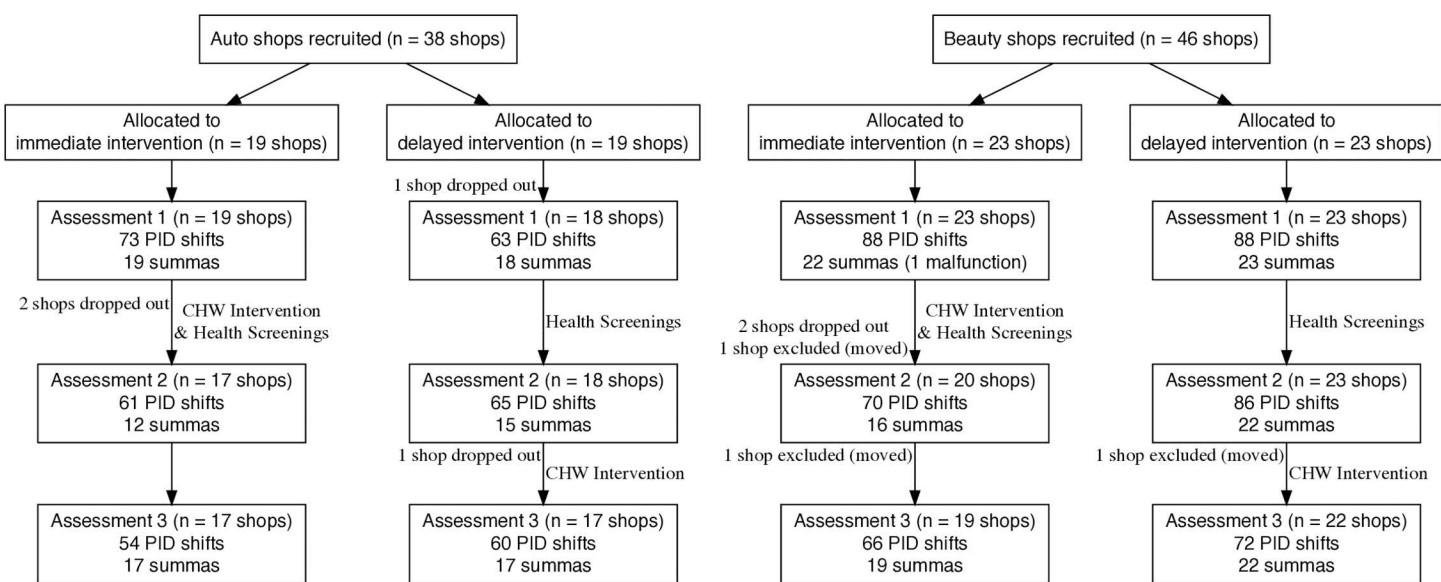

**Fig 1. Flow diagram of progress of clusters (shops) through phases of the randomized trial.** The diagram also shows the number of individual PID TVOC workshifts and the number of summa canisters (excluding duplicates and blanks) at each assessment in each group.

This trial was designed with immediate and delayed intervention groups, so that all participants could benefit from the intervention (which was an ethical issue for the partners), and to enhance study participation and engagement. A UA statistician created a computer-generated randomization list for each subtype of shop (Auto No Paint [ANP], Beauty Hair and Nails [BN], and Beauty Hair Only [BH]), to allocate shops to either the immediate or delayed intervention group and then sent these lists to CHWs at SERI at the start of the study. Because the intervention was often visible (e.g., a new air purifier), neither the participants nor the UA team could be guaranteed to remain blinded to the intervention status.

The trial began in February 2020 but was paused after a few initial assessments because of the COVID-19 pandemic. Before restarting in January 2022, several methodological changes were made to improve recruitment and reduce missing data. The initial design had two UA researchers observe participants throughout their workshifts [21]; however, because of pandemic-related hesitancy toward extra people in the shop, activities and ventilation conditions were self-reported using standardized logs. For each workshift, the UA assessment team would visit the shop at the start of the workshift to turn on and set up the monitors for logging. At this initial visit, the team provided participants with a brief tutorial on monitor operation, demonstrated troubleshooting procedures (e.g., how to restart the monitor by holding the center button when it alarmed with a flashing red light), and left contact information. The team would visit once in the middle of the workshift to check on the monitors and help participants complete missing information on the log, and then at the end of the workshift for final data and instrument collection. Participants were instructed to contact the UA team if the monitors turned off, stopped making their typical sampling noise (low hum), or there were other issues they could not resolve. During the exploratory aim to test the feasibility of monitoring and design the intervention, a tube was connected to the PID to measure TVOCs in a participant's breathing zone; however, the tube would frequently kink, resulting in no TVOCs being detected until a UA researcher noticed and fixed it [21]. Therefore, during the trial the tube was removed to minimize data loss. Additionally, the initial six-month timeline for completing three assessments per shop proved unrealistic, often extending longer (up to 21 months) because of scheduling challenges, temporary shop closures, and delays in the shop choosing controls.

The target sample size of 60 shops per sector was determined to provide 80% power to detect a 47% change in TVOC concentrations, which exceeded our assumed minimum important difference of 20%, but was less than the 59–98% reductions seen in specific VOCs when using an alternative spray gun cleaner [22]. Detailed calculation assumptions (including assumed intracluster correlation coefficient (ICC) values) are no longer available because of personnel changes. Observed ICCs were 0.32 for auto shops and 0.39 for beauty shops (see Results). Despite this target, recruiting 60 shops per sector (auto and beauty) from the designated ZIP codes was difficult because of the delayed start of the trial, so the study expanded to similar socioeconomic areas within the Tucson, AZ metro region, resulting in 38 auto and 46 beauty shops.

## Intervention

Before the intervention, the UA measurement team prepared product inventories by reviewing product labels, GHS safety data sheets (and for beauty salons, the EWG Skin Deep® Cosmetics Database), and entering these data into a SERI database for CHWs to use. For the intervention, CHWs assessed shop services, reviewed product inventories for VOC content, and suggested VOC exposure controls. CHWs reviewed possible controls with both employees and owners, with owners making final decisions on which to implement. Some controls, such as keeping containers closed when not in use, could be implemented immediately, whereas others required purchase (up to $300 provided per shop). Following the hierarchy of controls, CHWs tended to initially recommend product elimination or substitution, followed by engineering controls (e.g., air purifiers, trash cans with lids). Individual-level controls (e.g., PPE such as gloves) were suggested last according to the hierarchy, because they protect workers the least.

## Primary outcome: TVOCs

TVOCs were measured using a real-time PID ppbRAE 3000 (RAE Systems, Inc. San Jose, CA) either worn by participants on a belt or sling backpack or placed nearby. Detailed methods for these monitors, including calibration with

isobutylene, were previously published in a paper that reported on the exploratory study prior to the trial [21]. Because the PIDs were calibrated with isobutylene, they measured TVOCs in isobutylene equivalents.

The PIDs recorded TVOC measurements every 20 seconds for up to four workshifts at each shop for each of the three assessments. In shops with fewer than four workers, multiple workshifts for the same worker or workers were completed on different days to reach the four workshifts per assessment. Because these TVOC data were correlated in time, we used the time-weighted average (TWA) for each workshift, which is equivalent to the arithmetic mean because all time intervals were equal in length (20 s). The arithmetic mean was taken instead of the geometric mean even though these data are highly skewed, because the arithmetic mean gives a participant's overall exposure, if multiplied by the time they were exposed.

Before aggregation and analyses, the TVOC data were cleaned, and measurements below the limit of detection (LOD) of 1 ppb were imputed. Time periods where the ppbRAE was malfunctioning were identified by activity log notes (e.g., flow fault) and by data inconsistencies (e.g., if one worker's TVOC levels were entirely below the LOD while a nearby indoor worker had high levels for the whole workshift, it was assumed that the intake holes on the ppbRAE had been blocked for the work that was entirely below the LOD, and the data were invalid). Invalid data were removed. Measurements below the LOD, recorded as 0 ppb, were replaced with $\frac{LOD}{\sqrt{2}} = \frac{1}{\sqrt{2}} \approx 0.707$ [23]. Of the cleaned participant observations, 10.4% (99,333/957,322) were below the LOD. Once the data were aggregated by workshift as described above, 0.7% (6/846) of the data were below the LOD. Finally, because of their skewed distribution, these TVOC TWA data were log-transformed for statistical analysis.

### Secondary outcome: Hazard scores derived from specific VOCs

Concentrations of specific VOCs in the shop air over a workshift during an assessment were collected using Summa canisters (Restek™ SilcoCan Air Canisters with RAVE Valve). Summa data were attempted to be collected at all shops during Assessments 1 and 3, and at most shops during Assessment 2. Summa canisters were placed in the room with the highest expected activity in the shop in a location that would not interrupt the workflow. The Summa data were reported according to the US EPA Method TO-15, which tests for concentrations of over 70 VOCs. To ensure the reliability of our measurement methods, duplicate samples (i.e., two adjacent canisters) were taken in 6.8% (15/222) of the assessments, and blank samples were obtained in 5.9% (13/222) of them. Detailed methods have been provided in our prior paper [21].

Results for the chemicals that were detected in at least one shop (following US EPA Method TO-15) were plotted as a heatmap and used in the analysis. Variations in daily work practices may result in the detection of different specific VOCs with varying toxicities during different workshifts. Therefore, to enable meaningful comparisons, we established a workplace VOC hazard score. This score was calculated by dividing the concentration of each specific VOC by its reference value and summing these ratios for all measured VOCs in a Summa canister sample within a shop on a given date. A hazard score greater than one indicates a potential health risk from the VOC combination. Reference values, their sources, and the health effects of each specific VOC are presented in Table SM5 in S2 File. A comprehensive explanation of the hazard score calculation was reported previously [24].

### Potential covariates

**Outside TVOCs.** The UA team collected background measurements of the TVOCs in ambient air outside the shops. The average outside TVOCs for each shop assessment were calculated (and log-transformed for analysis because of their log-normal distribution), to serve as a potential covariate to account for other sources of VOCs in surrounding neighborhoods. Details are provided in section "Outside TVOCs" in S2 File.

**Air exchange rate.** To account for differences in ventilation in beauty shops, the air exchange rate (ACH) was calculated from indoor $CO_2$ concentrations measured at each assessment using an Aranet4 HOME sensor [25]. Beginning and end of day ACH values were averaged to get one air exchange rate for each shop at each assessment. Because of

their skewed distribution, air exchange rates were log-transformed for analysis. See section "Air exchange rate" in S2 File for details.

Air exchange rates were not calculated for auto shops, which usually have large garage doors open. Instead, to account for differences in ventilation in auto shops, the team collected shop-level data at each assessment on if participants worked outside or with open doors during their site audit visit, in addition to the self-reported ventilation data that participants recorded.

**Apparent air temperature.** Hourly apparent air temperature (wind chill or heat index) in Fahrenheit data from Tucson International Airport were downloaded from the Automated Surface Observing System network [26]. Then the average daytime (8 am – 8 pm) apparent air temperature for each date within the study range was calculated and matched to each workshift using the date. The apparent air temperature data were made missing for a date if hourly data for more than 9 of 13 hours was missing, which resulted in 1.4% (12/846) of workshifts missing apparent air temperature data. These data were considered for inclusion as a covariate at both the workshift level and shop level; the apparent air temperature at the shop level was the average over all the workshifts in a shop assessment.

**Other data collected.** CHWs at SERI created a FileMaker Pro database hosted by FMPHost to gather and organize control data, which they shared with UA researchers after study completion. The UA exposure assessment team collected and managed other shop and participant data using Research Electronic Data Capture (REDCap) hosted by UA [27,28]. This included shop characteristics from site audits at each assessment (including if participants were working outside or with open doors at the time), participant demographics, and participant self-reported activity and ventilation data. Additional information can be found in section "Other data collected" in S2 File.

## Statistical methods

Descriptive statistics were calculated for shop and participant characteristics. Differences in baseline characteristics between the immediate and delayed intervention groups were tested using the Welch Two-Sample t-test for continuous variables and Fisher's Exact Test for categorical variables. For this and subsequent analyses, all tests were two-sided, with a *p-value* < 0.05 considered statistically significant.

To analyze the primary outcome TVOC data, we used linear mixed models to account for the lack of independence from repeated measurements within a shop at an assessment and across assessments. We used contrasts to test the change in TVOC concentration from pre- to post-intervention between the arms. To account for possible confounding, we fit both unadjusted and adjusted models. The unadjusted model had log-transformed TVOC concentration as the outcome, with fixed effects for intervention group (immediate, delayed), assessment time (1, 2, 3), and their two-way interactions; random effects were included for shop and assessment time within shop, with random intercepts to account for correlations of workshifts within the same shop and the potentially stronger correlations of workshifts within the same shop at the same assessment. Random effects were assumed to be independent. Models for auto and beauty shops were run separately because of differences in potential covariates for adjusted models.

Covariates for the adjusted models presented here were selected based on their logical relevance and potential impact on exposures. For auto shops, additional covariates were baseline outside ventilation (i.e., working outside at Assessment 1, based on the UA exposure assessment team's site audit, because it could affect exposures and was imbalanced between arms at baseline) and average workshift apparent temperature (which could influence whether participants worked outside or kept garage doors open, as well as VOC levels). For beauty shops, additional covariates were baseline air exchange rate (because it could affect exposure) and whether the shop offered nail services (because shops providing both hair and nail services may have higher exposure). For both outside ventilation and air exchange rate, only baseline values were used in the models, because post-intervention values could be influenced by the intervention. The main results presented in this paper are from the adjusted models, because they provide a more accurate representation by accounting for relevant covariates.

To explore the impact of different model choices, additional adjusted model results are provided in S2 File. Specifically, covariates that changed the treatment effect by more than 10% when added one-at-a-time as a main effect to the unadjusted model for each sector were included. The following workshift-level covariates at each assessment were considered: apparent air temperature and self-reported activity and ventilation data. The following shop-level covariates at each assessment were also considered: apparent air temperature; outside TVOC concentration; if the building was rented or owned; if an owner or manager wore a monitor during that shop assessment; self-reported activity and ventilation data aggregated to the shop-level; if it was a home business; number of people working that day; number of rooms in shop; if the shop used "green" products; shop volume (calculated from measured shop dimensions); ventilation data (same categories as those for self-report plus wall heater, personal heater, central heater, and extractor); and for beauty shops only, air exchange rate and whether the beauty shop provided nail services in addition to hair.

To analyze the secondary outcome of hazard scores derived from specific VOC data, we used similar mixed models as described above to test the change in hazard score from pre- to post-intervention between the arms (although there was no random effect for workshift within a shop assessment, because workshift-level data were not taken for this outcome). Data preparation, cleaning, and statistical analyses were performed using *R* version 4.4.1 [29], the *tidyverse* package for data wrangling [30], the *lme4* package for linear mixed-effects models [31], and other packages listed in S2 File.

### Declaration of use of generative AI and AI-assisted technologies

During the preparation of this work the authors used ChatGPT and Claude to revise and edit the manuscript and to assist with R code for data analysis. After using these tools, the authors reviewed and edited the content and take full responsibility for the content of the published article.

## Results

### Recruitment, dates, and flow of shops through study

SERI CHWs conducted recruitment from January 10, 2022 to May 11, 2023. The UA team collected PID monitor TVOC data, Summa canister specific VOC data, and supplemental data (e.g., data on shop and participant characteristics) from January 15, 2022 to January 18, 2024. The trial was stopped before enrolling the originally planned 60 shops in each sector (auto or beauty), because of the reduced timeline caused by the COVID-19 pandemic and the increased difficulty of recruiting businesses in the post-pandemic period.

Fig 1 presents a flow diagram of the progress of shops through the phases of the randomized trial for auto and beauty shops. Only 6/84 (7%) of shops dropped out before completing the study. For auto shops, one shop dropped out before the first assessment, two shops dropped out between the first and second assessments, and one shop dropped out between the second and third assessments. For beauty shops, two shops dropped out between the first and second assessments. Reasons for dropping out included the owner no longer being interested or not wanting to continue during shop renovation; the shop was sold, and the new owner did not want to continue; the shop switched from an auto shop to a restaurant; and the shop closed. Three shops (4%) moved during the study, and their data after they moved were excluded from statistical analyses (but not from the heatmaps of specific VOCs, because those may remain of interest): one beauty shop moved between the first and second assessments (BH021), and two beauty shops moved between the second and third assessments (BH009 and BH028).

### Baseline shop and participant characteristics

Table 1 presents summaries of baseline shop characteristics by sector and intervention group. The only statistically significant difference was that 5.3% (1/19) of the auto shops in the immediate intervention group had outside ventilation (i.e., workers performing duties outdoors), compared to 39% (7/18) in the delayed intervention group (*p-value* = 0.02). Because

**Table 1. Baseline shop characteristics for auto and beauty shops that participated, by sector and intervention group.**

| | Auto | | | | Beauty | | | |
|---|---|---|---|---|---|---|---|---|
| | **Intervention group** | | | | **Intervention group** | | | |
| | **Immediate (N = 19)[1]** | **Delayed (N = 18)[1]** | **Total (N = 37)[1]** | **p-value** | **Immediate (N = 23)[1]** | **Delayed (N = 23)[1]** | **Total (N = 46)[1]** | **p-value** |
| **General business characteristics** | | | | | | | | |
| **Building rented or owned** | | | | 0.5[2] | | | | >0.9[2] |
| *Rented* | 12 (71%) | 9 (56%) | 21 (64%) | | 19 (83%) | 17 (81%) | 36 (82%) | |
| *Owned* | 5 (29%) | 7 (44%) | 12 (36%) | | 4 (17%) | 4 (19%) | 8 (18%) | |
| *Unknown* | 2 | 2 | 4 | | 0 | 2 | 2 | |
| **Home business** | 0 (0%) | 3 (17%) | 3 (8.1%) | 0.11[2] | 2 (10%) | 3 (16%) | 5 (13%) | 0.7[2] |
| *Unknown* | | | | | 3 | 4 | 7 | |
| **Uses green/eco-friendly products** | 10 (53%) | 5 (28%) | 15 (41%) | 0.2[2] | 12 (52%) | 13 (57%) | 25 (54%) | >0.9[2] |
| **Has unlabeled products** | 1 (5.3%) | 1 (5.6%) | 2 (5.4%) | >0.9[2] | 0 (0%) | 0 (0%) | 0 (0%) | >0.9[2] |
| **Workers trained to safely use products** | 19 (100%) | 16 (94%) | 35 (97%) | 0.5[2] | 23 (100%) | 23 (100%) | 46 (100%) | >0.9[2] |
| *Unknown* | 0 | 1 | 1 | | | | | |
| **Number people working today** | | | | 0.11[3] | | | | 0.6[3] |
| *Mean (SD)* | 4 (2) | 3 (2) | 3 (2) | | 3 (2) | 3 (2) | 3 (2) | |
| *Min – Max* | 1–8 | 1–8 | 1–8 | | 1–6 | 1–7 | 1–7 | |
| **Number of rooms in shop** | | | | 0.7[3] | | | | 0.4[3] |
| *Mean (SD)* | 4 (2) | 4 (1) | 4 (2) | | 4 (2) | 3 (1) | 3 (1) | |
| *Min – Max* | 2–10 | 1–7 | 1–10 | | 1–7 | 2–5 | 1–7 | |
| **Measured shop area (sq ft)** | | | | 0.9[3] | | | | 0.3[3] |
| *Mean (SD)* | 1,567 (1,066) | 1,645 (1,426) | 1,603 (1,228) | | 759 (309) | 666 (350) | 712 (330) | |
| *Min – Max* | 178–4,290 | 378–4,606 | 178–4,606 | | 196–1,476 | 91–1,372 | 91–1,476 | |
| *Unknown* | 2 | 3 | 5 | | | | | |
| **Measured shop height (ft)** | | | | 0.8[3] | | | | >0.9[3] |
| *Mean (SD)* | 13 (2) | 13 (3) | 13 (3) | | 9 (2) | 9 (1) | 9 (2) | |
| *Min – Max* | 9–17 | 8–17 | 8–17 | | 7–16 | 7–12 | 7–16 | |
| *Unknown* | 2 | 3 | 5 | | | | | |
| **Apparent temperature (F)** | | | | 0.2[3] | | | | 0.6[3] |
| *Mean (SD)* | 70 (17) | 76 (16) | 73 (17) | | 80 (14) | 78 (16) | 79 (15) | |
| *Min – Max* | 43–101 | 46–98 | 43–101 | | 43–97 | 49–98 | 43–98 | |
| **Outside TVOCs (ppb)** | | | | 0.2[3] | | | | 0.4[3] |
| *GM (GSD)* | 2 (4) | 5 (6) | 3 (5) | | 3 (5) | 2 (4) | 3 (4) | |
| *Median (Q1, Q3)* | 1 (1, 7) | 3 (1, 7) | 2 (1, 7) | | 3 (1, 5) | 1 (1, 4) | 2 (1, 5) | |
| *Min – Max* | 1–55 | 1–138 | 1–138 | | 1–75 | 1–164 | 1–164 | |
| *Unknown* | 4 | 1 | 5 | | 4 | 5 | 9 | |
| **Beauty shop type** | | | | | | | | >0.9[2] |
| *Hair and nails* | | | | | 6 (26%) | 5 (22%) | 11 (24%) | |
| *Hair only* | | | | | 17 (74%) | 18 (78%) | 35 (76%) | |

*(Continued)*

| | Auto | | | | Beauty | | | |
|---|---|---|---|---|---|---|---|---|
| | **Intervention group** | | | | **Intervention group** | | | |
| | **Immediate (N = 19)**[1] | **Delayed (N = 18)**[1] | **Total (N = 37)**[1] | **p-value** | **Immediate (N = 23)**[1] | **Delayed (N = 23)**[1] | **Total (N = 46)**[1] | **p-value** |
| **Air exchanges per hour** | | | | | | | | >0.9[3] |
| *GM (GSD)* | | | | | 2.06 (2.05) | 2.09 (2.34) | 2.08 (2.18) | |
| *Median (Q1, Q3)* | | | | | 1.95 (1.17, 3.23) | 1.98 (1.12, 3.75) | 1.97 (1.17, 3.23) | |
| *Min – Max* | | | | | 0.76–11.55 | 0.48–15.13 | 0.48–15.13 | |
| **Types of ventilation**[4] | | | | | | | | |
| **Central – window – door AC** | 5 (26%) | 5 (28%) | 10 (27%) | >0.9[2] | 21 (91%) | 16 (70%) | 37 (80%) | 0.13[2] |
| **Mini split** | 3 (16%) | 4 (22%) | 7 (19%) | 0.7[2] | 4 (17%) | 5 (22%) | 9 (20%) | >0.9[2] |
| **Swamp cooler** | 15 (79%) | 11 (61%) | 26 (70%) | 0.3[2] | 0 (0%) | 1 (4.3%) | 1 (2.2%) | >0.9[2] |
| **Fan** | 10 (53%) | 7 (39%) | 17 (46%) | 0.5[2] | 7 (30%) | 7 (30%) | 14 (30%) | >0.9[2] |
| **Local exhaust** | 1 (5.3%) | 0 (0%) | 1 (2.7%) | >0.9[2] | 2 (8.7%) | 0 (0%) | 2 (4.3%) | 0.5[2] |
| **Open door – window to outside** | 7 (37%) | 5 (28%) | 12 (32%) | 0.7[2] | 0 (0%) | 1 (4.3%) | 1 (2.2%) | >0.9[2] |
| **Outside** | 1 (5.3%) | 7 (39%) | 8 (22%) | **0.02**[2] | 0 (0%) | 0 (0%) | 0 (0%) | >0.9[2] |
| **Wall heater** | 0 (0%) | 0 (0%) | 0 (0%) | >0.9[2] | 0 (0%) | 0 (0%) | 0 (0%) | >0.9[2] |
| **Personal heater** | 0 (0%) | 0 (0%) | 0 (0%) | >0.9[2] | 0 (0%) | 0 (0%) | 0 (0%) | >0.9[2] |
| **Central heater** | 0 (0%) | 0 (0%) | 0 (0%) | >0.9[2] | 0 (0%) | 1 (4.3%) | 1 (2.2%) | >0.9[2] |
| **Other** | 1 (5.3%) | 1 (5.6%) | 2 (5.4%) | >0.9[2] | 1 (4.3%) | 3 (13%) | 4 (8.7%) | 0.6[2] |
| **Out of view** | 0 (0%) | 0 (0%) | 0 (0%) | >0.9[2] | 0 (0%) | 0 (0%) | 0 (0%) | >0.9[2] |
| **Local engineering controls**[4] | | | | | | | | |
| **Local exhaust ventilation** | 1 (5.3%) | 0 (0%) | 1 (2.7%) | >0.9[2] | 3 (13%) | 1 (4.3%) | 4 (8.7%) | 0.6[2] |
| **Extractor** | 2 (11%) | 2 (11%) | 4 (11%) | >0.9[2] | 4 (17%) | 1 (4.3%) | 5 (11%) | 0.3[2] |
| **Fan**[5] | 13 (68%) | 9 (50%) | 22 (59%) | 0.3[2] | 10 (43%) | 9 (39%) | 19 (41%) | >0.9[2] |
| **Air purifier** | 0 (0%) | 1 (5.6%) | 1 (2.7%) | 0.5[2] | 1 (4.3%) | 2 (8.7%) | 3 (6.5%) | >0.9[2] |
| **None** | 6 (32%) | 7 (39%) | 13 (35%) | 0.7[2] | 8 (35%) | 10 (43%) | 18 (39%) | 0.8[2] |

For variables that are log-normally distributed (air exchanges per hour and outside TVOCs), p-values were calculated on the log-transformed data. *P-values < 0.05 appear in bold.*

[1] Mean (SD) and range for continuous variables [GM (GSD), median (IQR), and range for log-normally distributed continuous variables outside TVOCs and air exchanges per hour]; n (%) for categorical variables; n for unknown. SD: standard deviation; GM: geometric mean; GSD: geometric standard deviation, IQR: interquartile range; TVOCs: total volatile organic compounds.

[2] Fisher's Exact Test for count data.

[3] Welch Two-Sample t-test.

[4] Percentages may sum to more than 100% within each section as shops could have multiple types of ventilation and multiple local engineering controls.

[5] Fan includes any of ceiling fan, box fan, desk fan, utility fan, portable fan, standing fan, or floor fan.

having outside ventilation (participants working outside) was imbalanced between the intervention groups at baseline, and because it may be associated with lower TVOC concentrations, this variable was added as a covariate in the adjusted model for auto shops.

Table 2 presents summaries of baseline demographics of participants by sector and intervention group. For each sector, there were no significant differences in demographics between intervention groups. Thus, no participant demographic variables were added as covariates in our adjusted model. Additionally, we recruited a majority of Latino

**Table 2. Baseline demographics for auto and beauty shop workers who used a PID monitor to measure TVOCs, by sector and intervention group.**

| | Auto | | | | Beauty | | | |
|---|---|---|---|---|---|---|---|---|
| | Intervention group | | Total (N = 121)[1] | p-value | Intervention group | | Total (N = 147)[1] | p-value |
| | Immediate (N = 69)[1] | Delayed (N = 52)[1] | | | Immediate (N = 77)[1] | Delayed (N = 70)[1] | | |
| **Gender** | | | | >0.9[2] | | | | >0.9[2] |
| *Male* | 64 (94%) | 48 (94%) | 112 (94%) | | 7 (9.3%) | 7 (10%) | 14 (9.7%) | |
| *Female* | 4 (5.9%) | 3 (5.9%) | 7 (5.9%) | | 68 (91%) | 62 (90%) | 130 (90%) | |
| *Unknown* | 1 | 1 | 2 | | 2 | 1 | 3 | |
| **Ethnicity**[4] | | | | 0.8[2] | | | | 0.6[2] |
| *Hispanic* | 47 (69%) | 37 (73%) | 84 (71%) | | 54 (72%) | 52 (78%) | 106 (75%) | |
| *Not Hispanic* | 21 (31%) | 14 (27%) | 35 (29%) | | 21 (28%) | 15 (22%) | 36 (25%) | |
| *Unknown* | 1 | 1 | 2 | | 2 | 3 | 5 | |
| **Employee type** | | | | 0.2[2] | | | | >0.9[2] |
| *Owner or Manager* | 17 (25%) | 18 (35%) | 35 (29%) | | 26 (35%) | 23 (33%) | 49 (34%) | |
| *Employee* | 51 (75%) | 33 (65%) | 84 (71%) | | 49 (65%) | 46 (67%) | 95 (66%) | |
| *Unknown* | 1 | 1 | 2 | | 2 | 1 | 3 | |
| **Has health insurance** | 39 (57%) | 30 (60%) | 69 (58%) | 0.9[2] | 45 (62%) | 49 (72%) | 94 (67%) | 0.2[2] |
| *Unknown* | 1 | 2 | 3 | | 4 | 2 | 6 | |
| **Age (years)** | | | | 0.3[3] | | | | 0.2[3] |
| *Mean (SD)* | 40 (12) | 38 (11) | 39 (12) | | 44 (12) | 47 (14) | 46 (13) | |
| *Min – Max* | 18–74 | 20–65 | 18–74 | | 25–69 | 21–72 | 21–72 | |
| *Unknown* | 5 | 11 | 16 | | 25 | 20 | 45 | |
| **Years worked in this field** | | | | >0.9[3] | | | | 0.2[3] |
| *Mean (SD)* | 16 (13) | 16 (14) | 16 (13) | | 19 (11) | 21 (14) | 20 (12) | |
| *Min – Max* | 0–43 | 0–54 | 0–54 | | 0–42 | 1–52 | 0–52 | |
| *Unknown* | 1 | 1 | 2 | | 2 | 1 | 3 | |
| **Years worked in this shop** | | | | 0.4[3] | | | | >0.9[3] |
| *Mean (SD)* | 7 (9) | 6 (6) | 6 (8) | | 7 (7) | 7 (6) | 7 (7) | |
| *Min – Max* | 0–37 | 0–22 | 0–37 | | 0–32 | 0–25 | 0–32 | |
| *Unknown* | 1 | 1 | 2 | | 2 | 1 | 3 | |
| **Usual length of workshift (hours)** | | | | 0.4[3] | | | | >0.9[3] |
| *Mean (SD)* | 9 (1) | 9 (2) | 9 (1) | | 8 (2) | 8 (2) | 8 (2) | |
| *Min – Max* | 5–12 | 6–14 | 5–14 | | 4–12 | 4–12 | 4–12 | |
| *Unknown* | 2 | 1 | 3 | | 4 | 1 | 5 | |
| **Total days/week usually worked** | | | | 0.6[3] | | | | 0.14[3] |
| *Mean (SD)* | 5 (1) | 5 (1) | 5 (1) | | 5 (1) | 5 (1) | 5 (1) | |
| *Min – Max* | 4–6 | 4–7 | 4–7 | | 1–7 | 2–7 | 1–7 | |
| *Unknown* | 1 | 1 | 2 | | 2 | 1 | 3 | |

*P-values < 0.05 would appear in bold.*

[1]Mean (SD) and range for continuous variables; n (%) for categorical variables; n for unknown.

[2]Fisher's Exact Test for count data.

[3]Welch Two-Sample t-test.

[4]Ethnicity was asked as Hispanic/Not Hispanic. However, many Hispanic/Latino participants did not relate to this question and would first answer something along the lines of "I am Mexican, from ___ part of Mexico" before finally choosing "Hispanic." Latino has been used in the body of the manuscript, because it seems the more appropriate term for this group of people with origins in Latin America.

participants: 71% of auto shop participants and 75% of beauty shop participants who answered the question about ethnicity were Latino.

### Intervention: Controls chosen

Table SM1 in S2 File presents data from SERI on the controls shops agreed to or were already doing, by sector and intervention group. Of the program-purchased controls, most auto shops (61%) chose PPE, followed by fans and new equipment (55% each); however, most beauty shops (87%) chose air purifiers, followed by trash cans with lids and PPE (65% each). Although CHWs suggested PPE last according to the hierarchy of controls, the community-based approach required honoring owner requests for PPE appropriate to the specific tasks and products, which addressed occupational safety needs extending beyond VOC exposure alone. Additionally, only 3% of auto shops agreed to eliminate products (vs. 15% for beauty), and 3% of auto shops developed new policies (vs. 13% for beauty). The controls were generally well balanced between intervention groups within a sector. However, more auto shops in the immediate intervention group already used controls such as purchasing smaller containers to minimize waste and keeping lids on solvent parts washers when not in use. Because the immediate intervention group had higher levels of TVOCs on average than the delayed intervention group did at baseline, these differences were not thought to significantly affect the study results.

### Primary outcome: TVOCs

The TWA TVOC concentrations for each of the 846 workshifts (376 auto; 470 beauty) at 236 shop assessments (106 auto; 130 beauty) were analyzed. Fig 2 shows the TWA TVOC concentrations for each workshift for each shop at each assessment, colored by sector and intervention group. For an aggregated view of the data, Fig SM1 in S2 File shows violin plots of average TVOCs at each shop at each assessment for each intervention group. The substantial unexplained variability in TVOC concentrations – both within individual shops at an assessment and between different shops – meant that our statistical analysis could not detect a clear intervention effect.

For auto shops, the adjusted model for TVOCs included the covariates of shop-level baseline outside ventilation (i.e., working outside) and workshift-level apparent temperature. For beauty shops, the adjusted model included shop-level log-transformed baseline air exchange rate and whether the shop provided nail services. Table SM2 in S2 File provides the specifications for each model, including the alternate (variant) adjusted models.

The intervention did not statistically significantly reduce TVOCs for either auto or beauty shops. Specifically, Table 3 shows that, on average, for the adjusted model for auto shops after the intervention, there was a 28% non-statistically significant increase in TVOCs, with a 95% CI ranging from a 46% reduction to a 203% increase in TVOCs. On average, for the adjusted model for beauty shops after intervention, there was a 27% non-statistically significant reduction in TVOCs, with a 95% CI ranging from a 55% reduction to a 19% increase in TVOCs. The effect of the intervention was also not statistically significant for the unadjusted and alternate adjusted models presented in Table SM3 and Fig SM2 in S2 File.

The large unexplained variability in the TVOC data made it difficult to detect a statistically significant intervention effect, because it reduced the statistical power of the analysis. In the adjusted models, roughly half of the variance (55% for auto shops; 46% for beauty shops) was not accounted for by the model. The shop-level intracluster correlation coefficient (ICC), representing the proportion of total variance attributable to between-shop differences, was 0.32 for auto shops and 0.39 for beauty shops (Table SM4 in S2 File). Additionally, the total variance in log(TVOC) for auto shops (3.5) was greater than that for beauty shops (1.5; Table SM4 in S2 File). The larger fraction of unexplained variance and the larger total variance in auto shops resulted in larger 95% CIs for the intervention effect in auto shops than in beauty shops.

When estimated TVOC concentrations in different sectors were compared via adjusted models, beauty shop exposures were an order of magnitude higher than those in auto shops: GM (95% CI) = 3.1 (2.3, 4.2) ppm compared to 0.22 (0.13, 0.36) ppm (see Table SM3 in S2 File). Additionally, hair-and-nail beauty shop exposures were approximately 3 times higher than those in hair-only beauty shops: 5.3 (3.1, 9.0) ppm compared to 1.8 (1.3, 2.4) ppm.

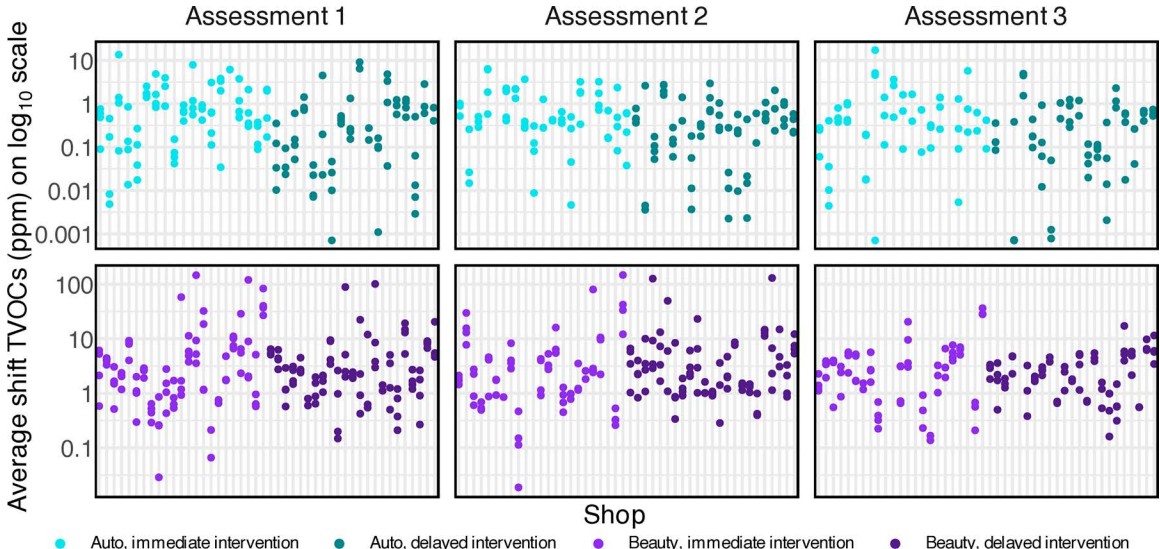

**Fig 2. TWA TVOC concentrations for each workshift for each shop at each assessment.** Data for each workshift at a shop have been placed on the same vertical line. Auto shop data are shown in the top three plots, and beauty shop data are shown in the bottom three plots. Data from shops in the immediate intervention group are on the left side of each plot, and data from shops in the delayed intervention group are on the right.

**Table 3. Estimated intervention effects along with their 95% CIs from the unadjusted and adjusted mixed models for auto and beauty shops.**

| Model | Est. intervention effect (95% CI) |
|---|---|
| Auto Unadjusted | 0.86 (0.35, 2.13) |
| Auto Adjusted | 1.28 (0.54, 3.03) |
| Beauty Unadjusted | 0.74 (0.46, 1.20) |
| Beauty Adjusted | 0.73 (0.45, 1.19) |

Results have been back-transformed to the original scale from the log scale, which results in asymmetrical CIs and differences on the log scale have become ratios on the original scale. The estimated intervention effect should be evaluated by comparing to 1: a value less than 1 indicates that the intervention was effective in reducing TVOCs, a value greater than 1 indicates that the intervention may have increased TVOCs, and a 95% CI that includes 1 indicates that the effect of intervention was not statistically significant (so we cannot determine if it had a beneficial effect or not). Est.: Estimated; CI: Confidence interval.

### Secondary outcome: Hazard scores derived from specific VOCs

Overall the quality control results were acceptable for duplicate and blank samples. No blank corrections were necessary (section "Quality control data for specific VOCs" in S2 File).

The intervention did not statistically significantly reduce hazard scores for either auto or beauty shops, with even wider 95% CIs for intervention effects than for TVOCs (section "Mixed models for hazard scores" in S2 File). However, we used adjusted models to compare estimated hazard scores between sectors. Auto shops had similar estimated hazard scores to those of beauty shops: GM (95% CI) = 0.30 (0.14, 0.64) for auto shops compared to 0.38 (0.21, 0.70) for beauty. Additionally, hair-and-nail beauty shops had a factor of ~6 higher estimated hazard scores than did hair-only beauty shops: GM (95% CI) = 0.97 (0.33, 2.80) compared to 0.15 (0.08, 0.27).

Approximately one-third of all assessments had hazard scores greater than one, indicating that the shop air at these assessments was a potential hazard to human health. Specifically, the hazard scores for all auto shop assessments in this

study ranged from 0.0006 to 89, and 39/98 (40%) of the shop assessments had hazard scores greater than 1. Additionally, the hazard scores for all the beauty shop assessments in this study ranged from 0.004 to 34, and 38/124 (31%) of the shop assessments had hazard scores greater than 1.

Figs SM4 and SM5 in S2 File present heatmaps of specific VOC concentrations measured using the US EPA TO-15 method, along with the calculated hazard score, for auto and beauty shops, respectively. The specific VOCs detected in at least half of the 37 auto shops with Summa canisters (with the number of shops each was detected in) were as follows: ethanol (37), toluene (36), acetone (34), heptane (34), benzene (32), 2,2,4-trimethylpentane (30), hexane (30), ethylbenzene (29), m&p-xylene (29), o-xylene (29), cyclohexane (25), 1,2,4-trimethylbenzene (24), and methylcyclohexane (24). The specific VOCs detected in at least half of the 46 beauty shops (with the number of shops each was detected in) were as follows: acetone (46), ethanol (46), 2-propanol (IPA) (45), toluene (39), ethyl acetate (34), and heptane (25). In auto shops, the following specific VOCs tended to drive high hazard scores: m&p-xylene, benzene, 1,2,4-trimethylbenzene, hexane, and acetone; these VOCs are likely a direct result of the products and processes in use, ranging from fuels to solvents used in auto maintenance and repair work. In beauty shops, the following specific VOCs tended to drive high hazard scores: acetone, chloroform, 1,2-dichloroethane, tetrachloroethene, and trichloroethene; whereas acetone is commonly used in beauty products, the chlorinated solvents may not reflect beauty shop products and processes but rather contributions from external sources, such as adjacent dry cleaners.

To observe the relationship between our secondary and primary outcomes, we plotted hazard scores versus TVOCs, each on the $\log_{10}$ scale (Fig 3). Shop TVOC concentrations were directly correlated with hazard scores on the $\log_{10}$ scale, with $R^2$ ranging from 0.22 to 0.41, indicating that only 22–41% of the variation in $\log_{10}$(hazard score) was explained by $\log_{10}$(TVOCs). Whereas beauty shops tended to have higher TVOCs than auto shops did, they had a similar range of hazard scores (most shop assessments had hazard scores between 0.01 and 10). Additionally, the slopes of the regression line fits for the different subsets of the data were all near 1, as expected (if we assume that the hazard score is proportional to TVOCs, then the slope on the $\log_{10} - \log_{10}$ plot should be 1; see section "Relationship between hazard scores and TVOCs and comparison to Ricklund et al. (2022)" in S2 File).

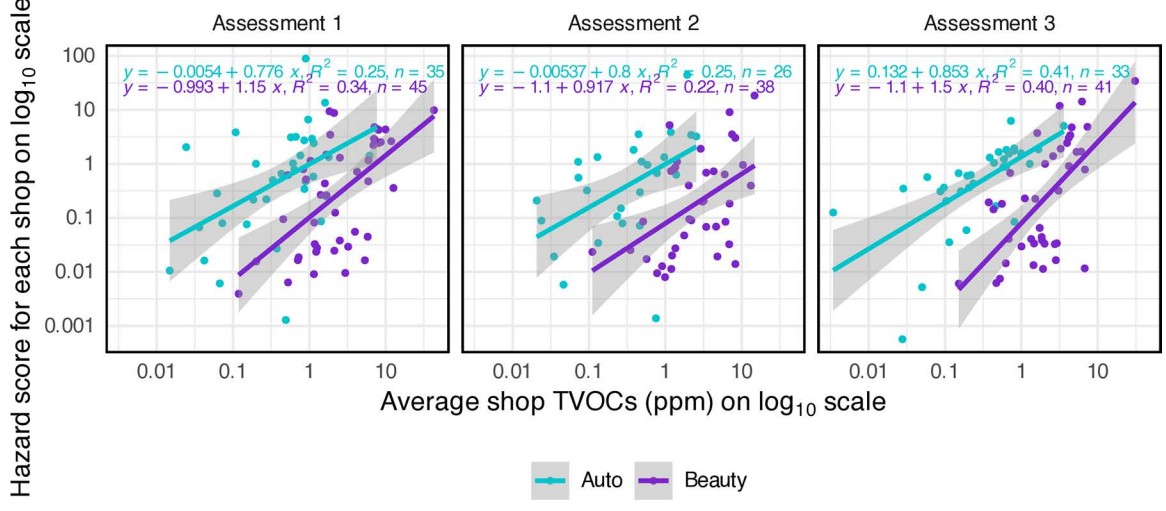

**Fig 3. Hazard scores versus average shop TVOC concentrations at each assessment.** The average shop TVOC concentrations were calculated by taking the geometric mean of the TWA TVOCs for each workshift of that shop for that assessment. Regression lines with 95% CIs are shown for each sector at each assessment.

## Outside TVOCs

For these auto and beauty shops, outside TVOC concentrations (GM: 3 ppb; Q1: 1 ppb; Q3: 6 ppb) were generally much lower than the TVOC concentrations measured during the shop assessments (GM: 906 ppb; Q1: 432 ppb; Q3: 2,768 ppb). Consequently, outside TVOCs were not significantly correlated with TVOCs during a shop assessment (Fig SM9 in S2 File). As a result, they were not used in the analyses.

## Air exchange rates

To understand the relationship between the air exchange rate and our primary outcome in beauty shops, we plotted TVOCs versus the air exchange rate in air changes per hour (ACH), each on the $\log_{10}$ scale (Fig 4). Higher air exchange rates were loosely correlated with lower average TVOCs, with $R^2$ values ranging from <0.01 to 0.31, indicating that up to 31% of the variation in $\log_{10}$(TVOCs) was explained by $\log_{10}$(ACH).

The air exchange rates in beauty shops in this study were generally lower than the minimum 4–6 ACH recommended for small-volume indoor retail spaces such as beauty shops [32] and the 5.5 ACH calculated from the ASHRAE [33] minimum recommended ventilation for beauty shops specifically (0.83 cfm/ft$^2$) using our mean beauty shop ceiling height (9 ft): $\left(\frac{0.83 \text{ ft}^3}{(\text{min} \cdot \text{ft}^2)}\right) \cdot \left(\frac{60 \text{ min}}{\text{hr}}\right) \cdot \left(\frac{1}{9 \text{ ft}}\right) = 5.5$ ACH. At baseline, the air exchange rates in the beauty shops in this study ranged from 0.5 to 15.1 ACH, with a geometric mean of 2.1 ACH, and 83% (38/46) of the shops had rates less than 4 ACH. For all the beauty shop assessments in this study, the air exchange rates ranged from 0.4 to 17.8 ACH, with a geometric mean of 1.9 ACH, and 87% (112/129) of the shop assessments had rates less than 4 ACH. We did not observe a significant change in the air exchange rates in the shops from the intervention. However, the air purifiers provided by the program as VOC-reduction controls would not affect the $CO_2$ concentrations in the shop and therefore would not affect the estimated air exchange rates.

## Discussion

This community-engaged cluster randomized controlled trial evaluated whether a CHW-led industrial hygiene intervention could reduce workplace VOC exposure in small businesses that mainly employ marginalized workers. In both auto repair

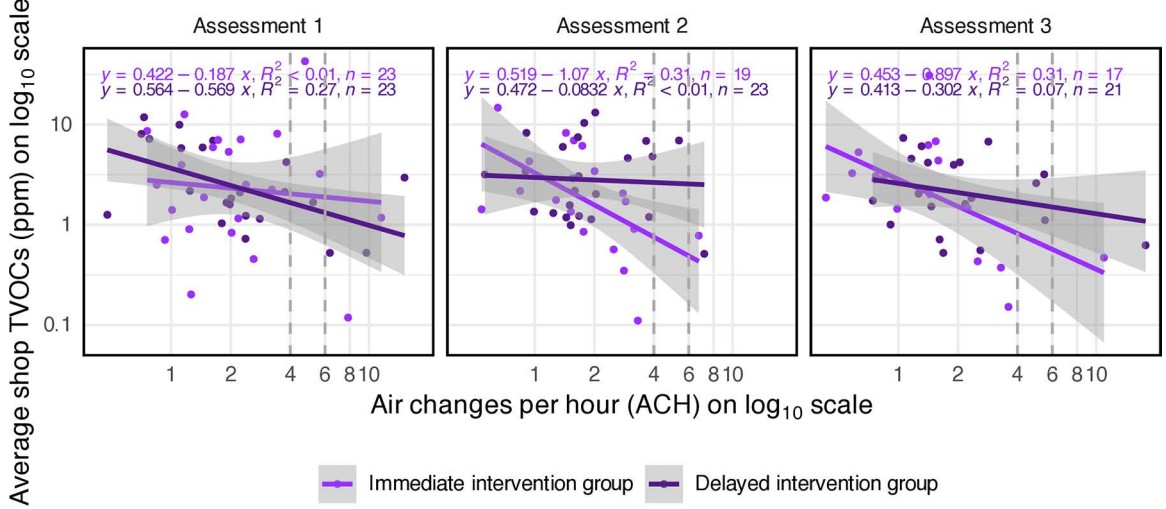

**Fig 4. Average shop TVOCs versus air exchange rate for beauty shops.** Each datapoint in this plot represents TVOC and air exchange rate data at one beauty shop at each assessment: shop TVOCs are the geometric average of the TWA TVOCs for each workshift of that shop for each assessment; air exchange rates are the average of the start- and end-of-day air exchange rates at each shop assessment. Regression lines with 95% CIs are shown for each group. Grey vertical dashed lines at the minimum recommended 4-6 ACH for beauty shops are shown for reference.

and beauty shops, there was no statistically significant reduction in TVOCs or hazard scores from the intervention; however, in beauty shops the point estimate indicated a 27% reduction in TVOCs, demonstrating that this intervention has the potential to reduce TVOC exposure and should be explored further. CHWs' strong community connections ensured high retention (93% of shops), demonstrating their effectiveness in engaging this marginalized population. Additionally, CHWs collaborated with participants to identify VOC exposure controls acceptable to auto repair and beauty shop workers. Finally, this study gathered valuable VOC exposure and accompanying shop and participant data for many small business auto repair and beauty shops over several assessments. Studies with high-quality VOC data in these specific occupational settings are rare, and consistent measurement methods enabled the comparison of VOC exposures between sectors. The TVOC concentrations in beauty shops were an order of magnitude higher than those in auto shops, with inadequate ventilation identified as a key issue.

The intervention's lack of statistically significant effects likely stemmed from unexpectedly large variability in VOC data, overestimated intervention impact, and recruitment challenges that reduced statistical power. High VOC variability (influenced by personnel turnover, product formulation changes, and seasonal shifts) complicated outcome measurements. Few comparable studies exist, but prior research on reducing TVOCs in nail salons using a different community-based approach (without a control group) similarly found no significant reductions, likely because of large variations in their TVOC data caused by practice variations [16]. Given the large variability in our data, an average 20% reduction in TVOCs (which we consider to be the minimum meaningful change) is plausible for beauty shops.

The collaborative approach through which CHWs engaged employees to help owners select program-purchased controls provided insights into controls that were acceptable to auto and beauty shop workers. The results appeared more promising for beauty shops, where the point estimate suggested a 27% average reduction in TVOCs. This may be because beauty shops more frequently chose controls that directly affected VOC concentrations in air (e.g., air purifiers, lidded trash cans), whereas auto shops favored PPE, which may reduce worker exposure but not airborne VOC concentrations. Furthermore, controls selected by the beauty shops were more likely to be effective according to the hierarchy of controls [34]. Other potential reasons why the intervention was more promising for beauty than auto shops include (1) more effective controls for auto shops were too expensive (>$300/shop budget; e.g., a paint booth would cost > $10,000), (2) beauty shops were more willing to adhere to the full study protocol (4% of beauty shops dropped out vs. 11% of auto shops), and (3) beauty shop participants may have considered the health risks to their clients as well as to themselves [35]. Given the evidence on the effectiveness of air purifiers and the preference of the beauty shops for this intervention, future implementation studies could provide air purifiers and monitor their use.

Despite having similar ranges of hazard scores, beauty shops had TVOC concentrations that were approximately an order of magnitude higher than those in auto repair shops, suggesting greater inhalation risks from VOCs. This disparity suggests a gap in research: many VOCs commonly present in beauty shops have not been studied as thoroughly as those present in auto shops, leaving the specific health risks associated with beauty shop VOCs less well understood. Compared to hair-only beauty shops, beauty shops offering both hair and nail services had approximately three times higher TVOC concentrations and six times higher hazard scores. This outcome is expected, given the well-documented risks associated with nail treatments that use quickly evaporating solvents such as toluene and acetone [16–18].

While neither auto shop model showed a statistically significant intervention effect, the direction of the point estimate of the intervention effect changed after adjustment: it went from a 14% reduction for our unadjusted model to a 28% increase for our adjusted model (Table 3). This was primarily caused by including the covariate of apparent temperature, which we selected because it can influence behaviors (e.g., working outside or keeping garage doors open) and TVOC levels. Its inclusion in the model changed the direction of the estimated intervention effect from negative to positive (numerically by 48%, well above our 10% threshold for covariate inclusion) and improved model fit (cAIC difference −4; threshold < −2). The reason for this directional shift is unclear. It may be because point estimates for our intervention effect are sensitive to model specification, in part because, despite this being a rich occupational-exposure dataset, the effective sample

size remains relatively small. In general, increasing apparent temperature was associated with decreased TVOCs (likely because shops were more likely to have their doors open and work outside with increasing temperature), except in the delayed intervention group during Assessment 2, where TVOCs increased with apparent temperature (see Fig SM10 in S2 File). Another anomaly at Assessment 2 was that only higher-than-average apparent temperatures were measured in the immediate intervention group. These anomalies at Assessment 2 likely reflect small-sample variability that contributed to this shift in the point estimate. Even so, apparent temperature remains a relevant covariate warranting further investigation. Importantly, regardless of model choice, the overall conclusion holds: the estimated impact of intervention in auto shops is associated with broad confidence intervals and is not associated with statistically significant results.

The air exchange rates in beauty shops, which ranged from 0.43 to 17.77 ACH, were unacceptably low, with 87% of the assessments falling below the minimum of 4 ACH. Published data on beauty shop air exchange rates are limited, but the lowest rates reported here match those reported (0.28–0.41 ACH) in a Spanish study [36]. Improving ventilation is essential, requiring both individual actions (e.g., opening windows) and systemic policy changes (e.g., stricter building codes). However, improving air quality in beauty shops, especially in marginalized communities, must address systemic inequalities, as access to clean air is often influenced by wealth and social status, leaving marginalized groups in polluted, poorly ventilated spaces [37]. Recognized as a fundamental human right by the World Health Organization, clean indoor air requires a renewed focus on ventilation as a public health priority [38]. Our CHWs have been sharing these findings with government agencies and community groups in Tucson, mirroring advocacy efforts that led to ventilation standards for nail salons in New York City [39].

The TVOC values reported in the literature are often not comparable, because the methods used to measure and define TVOCs are inconsistent [40]. Although no comparable data exist for auto shops, beauty shop studies provide some reference points. TVOC concentrations in five UK Afro-Caribbean salons [41] ranged from 366 to 8,216 ppb, within our observed range (19–147,871 ppb). Additionally, those in 62 Taiwanese salons [42] of 75.3 ± 47.2 ppb were much lower than the GM for our beauty shops of 3,085 ppb with 95% CI (2,277, 4,179) ppb, potentially due to differences in monitoring duration, products, or ventilation. A review of 23 studies of airborne hazardous chemicals (including TVOCs and specific VOCs) in hair salons highlighted substantial variability in airborne chemical concentrations both within and among individual studies, reinforcing the challenges of cross-study comparisons [43].

Both TVOCs and hazard scores were measured in ten Swedish hair salons using a different method than that used in our study [44]. Their TVOC concentrations were roughly an order of magnitude lower than those in our beauty shops when converted to the same units (ppb isobutylene equivalents), likely because of differences in methods, reference values, products, or ventilation. Despite this, both studies had similar ranges of hazard scores (mostly within 0.01–10), and the regression slopes for $\log_{10}$(hazard score) on $\log_{10}$(TVOCs) were all near 1, as expected. The value of $R^2$ for this regression from their study (0.41) was also comparable to ours (0.22–0.40). These similarities suggest that the relationship between TVOCs and hazard scores may hold for different methods and settings, where direct TVOC comparisons may not.

## Strengths

Key strengths of this study were its assiduous cluster-randomized controlled trial design and its community-engaged research approach. Such a rigorous trial design is rare in the field: a systematic review of CHWs' role in occupational safety and health research revealed that only one out of 17 studies used a true experimental design [45]. The community partnership between UA and SERI was essential for this community-engaged research: SERI's involvement was essential for designing, implementing, and evaluating VOC exposure controls for marginalized workers at these small businesses, and their deep community ties enhanced retention (93%).

Additional strengths included the use of low-impact measurement techniques, the ability to compare TVOCs between business sectors, and air exchange rate measurements in many beauty shops. The low-impact monitoring approaches for TVOCs and specific VOCs previously piloted by our group allowed us to capture inhalation VOC exposures with minimal disruption to shop workers and their clients. Consistent VOC measurement methods in different business sectors allowed

us to compare inhalation VOC exposure risks between them. Finally, measuring air exchange rates in 46 beauty shops enabled us to highlight the widespread issue of inadequate ventilation in these spaces.

## Limitations

The most notable limitation was recruiting only approximately 40 of the targeted 60 shops per sector, which reduced the power to assess the intervention effect. Second, the PIDs did not directly sample TVOCs in the participants' breathing zone, and for practical reasons, the participants sometimes placed the PID nearby instead of wearing it, potentially leading to measurement inaccuracies; however, this was consistent between arms and likely did not affect the assessment of the intervention. Additionally, the secondary outcome of hazard scores had three limitations that led it to often underestimate exposure risk: (1) the Summa canisters could measure only ~70 of the thousands of VOCs that may be present in these workplaces; (2) not every specific VOC measured by the Summa canisters had a reference value; and (3) the Summa canisters were generally not as close to the participants and the solvents they were using as the PIDs.

This study also had potential sources of bias. Mainly, having the UA exposure assessment team conduct a site audit at each visit (where they asked questions about ventilation types and engineering controls) was a potential source of bias, because the team could infer the controls and might judge their effectiveness, which might influence them when they went to take future VOC measurements at other shops. Additionally, having the UA team at shops about three times each work-shift could have biased the results, because seeing them served as a reminder to the participants to reduce VOCs. For example, when the UA team arrived, a participant would prop open a window or turn on a fan.

## Conclusions

While this CHW-led industrial hygiene intervention did not yield statistically significant reductions in TVOCs in the air of auto repair or beauty shops, a meaningful 20% average reduction in TVOCs remains plausible, especially for beauty shops. CHWs, with their deep community connections, achieved strong retention rates and identified acceptable VOC exposure controls for auto and beauty shop workers, suggesting that future studies should explore acceptable controls such as air purifiers or increased ventilation in beauty shops. The study highlighted key findings: (1) beauty shops may be more dangerous places to work than auto shops in terms of exposure to VOCs; (2) beauty shops exhibit unhealthily low air exchange rates, indicating a need for greater awareness and potential policy action to improve ventilation standards; and (3) the feasibility of a scalable, community-engaged intervention – delivered for about $300 per shop – demonstrates a promising model for protecting vulnerable workers. Future interventions should build upon this community-based approach with enhanced VOC measurement methodology, larger sample sizes to achieve adequate statistical power, and graduated implementation steps to more effectively reduce occupational health disparities in small businesses.

## Supporting information

**S1 Checklist. CONSORT checklist for cluster randomized trials.**
(PDF)

**S2 File. The main Supplemental Material file with all supplemental figures and tables referred to in the body of the manuscript.**
(PDF)

## Acknowledgments

Thank you to the participants at auto and beauty shops in metropolitan Tucson, AZ. This study would not have been possible without their cooperation. Thanks to Catherine Ornelas for assisting with the data collection. Thank you also to Dr. Niklas Ricklund for sharing the TVOC and hazard index data from his paper on Swedish hairdressers' occupational exposure to VOCs, so that we could directly compare them with our data.

## Author contributions

**Conceptualization:** Maia Ingram, Scott Carvajal, Ann Marie Wolf, Paloma I. Beamer.

**Data curation:** Shannon L. Gutenkunst, Carolina Quijada, Marvin Chaires, Emma V. Gallardo, Jenna K. Honan, Karla Bayless, Xavier Chaidez, Jacqueline L. Larson, Ann Marie Wolf.

**Formal analysis:** Shannon L. Gutenkunst, Dean Billheimer.

**Funding acquisition:** Paloma I. Beamer.

**Investigation:** Carolina Quijada, Marvin Chaires, Imelda Cortez, Flor Sandoval, Jella Balgos, Emma V. Gallardo, Pedro Flores Gallardo, Sam Sneed, Jenna K. Honan, Karla Bayless, Xavier Chaidez, Cristobal Reyes Cuevas, Jacqueline L. Larson, Fernanda J. Camargo, Denise Moreno Ramírez, Maia Ingram, Paloma I. Beamer.

**Methodology:** Nathan Lothrop, Carolina Quijada, Marvin Chaires, Sam Sneed, Jenna K. Honan, Jacqueline L. Larson, Paloma I. Beamer.

**Project administration:** Carolina Quijada, Marvin Chaires, Flor Sandoval, Emma V. Gallardo, Ann Marie Wolf, Paloma I. Beamer.

**Resources:** Dean Billheimer, Ann Marie Wolf, Paloma I. Beamer.

**Software:** Shannon L. Gutenkunst, Dean Billheimer.

**Supervision:** Maia Ingram, Dean Billheimer, Ann Marie Wolf, Paloma I. Beamer.

**Validation:** Shannon L. Gutenkunst.

**Visualization:** Shannon L. Gutenkunst.

**Writing – original draft:** Shannon L. Gutenkunst.

**Writing – review & editing:** Shannon L. Gutenkunst, Nathan Lothrop, Carolina Quijada, Marvin Chaires, Emma V. Gallardo, Pedro Flores Gallardo, Sam Sneed, Xavier Chaidez, Maia Ingram, Dean Billheimer, Ann Marie Wolf, Paloma I. Beamer.

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
