## [Decision Letter · Decision Letter 0]

29 Jan 2026

PONE-D-25-61403Community health worker intervention to reduce worker exposure to volatile organic compounds in small business auto and beauty shops in a marginalized community: a cluster randomized controlled trialPLOS One

Dear Dr. Gutenkunst,

Thank you for submitting your manuscript to PLOS ONE. After careful consideration, we feel that it has merit but does not fully meet PLOS ONE’s publication criteria as it currently stands. Therefore, we invite you to submit a revised version of the manuscript that addresses the points raised during the review process.

We look forward to receiving your revised manuscript.

Kind regards,

Rajeev Singh

Academic Editor

PLOS One

Journal Requirements:

[This project was supported by the National Institute of Environmental Health Sciences grants R01 ES028250, P30 ES006694, T32 ES007091, and R25 ES025494. The publication’s contents are solely the authors’ responsibility and do not necessarily represent the official views of the National Institutes of Health.].

3. Thank you for stating the following in your manuscript:

[This project was supported by the National Institute of Environmental Health Sciences grants R01 ES028250, P30 ES006694, T32 ES007091, and R25 ES025494. The publication’s contents are solely the authors’ responsibility and do not necessarily represent the official views of the National Institutes of Health.]

[This project was supported by the National Institute of Environmental Health Sciences grants R01 ES028250, P30 ES006694, T32 ES007091, and R25 ES025494. The publication’s contents are solely the authors’ responsibility and do not necessarily represent the official views of the National Institutes of Health.]

4. In the online submission form you indicate that your data is not available for proprietary reasons and have provided a contact point for accessing this data. Please note that your current contact point is a co-author on this manuscript. According to our Data Policy, the contact point must not be an author on the manuscript and must be an institutional contact, ideally not an individual. Please revise your data statement to a non-author institutional point of contact, such as a data access or ethics committee, and send this to us via return email. Please also include contact information for the third-party organization, and please include the full citation of where the data can be found.

6. We note that you have uploaded Supporting Information figures/tables that were not cited in your manuscript. Please update any in-text citations for your Supporting Information files in your manuscript. Please see our Supporting Information guidelines for more information: http://journals.plos.org/plosone/s/supporting-information.

Reviewers' comments:

Reviewer's Responses to Questions

**Comments to the Author**

1. Is the manuscript technically sound, and do the data support the conclusions?

Reviewer #1: Partly

Reviewer #2: Yes

2. Has the statistical analysis been performed appropriately and rigorously?

Reviewer #1: Yes

Reviewer #2: Yes

3. Have the authors made all data underlying the findings in their manuscript fully available?

Reviewer #1: Yes

Reviewer #2: Yes

4. Is the manuscript presented in an intelligible fashion and written in standard English?

Reviewer #1: Yes

Reviewer #2: Yes

5. Review Comments to the Author

Reviewer #1: Firstly, this was a great study to support the marginalized workers with approaches to providing occupational health guidance

However, I have the following questions or request for clarity on the following:

Overall - the study has many limitations regarding methodology and sample size subjecting it not to make strong conclusion about the outcome. The equipment used in this study also had few limitations including placing them at the measurement sites ensuring proper sampling technique.

The study compared auto repair shops and beauty salons. However, it is not mentioning about reviewing the GHS safety data sheets of chemicals used at these two different facilities. Instead, the study checked if the chemical were unlabelled or not, but no information contained in the label was reviewed.

Not clear if participants were trained to look after the samplers before the study could take off since the researchers only visited three times to check and assist with questionnaires. It was not confirmed if the CHW had training to perform these measurements, equipment operation and they would be in a position to apply professional judgement similar to if it was done by the Industrial Hygienist for proposing industrial l hygiene interventions.

Line 320 referred to Supplementary material on PPE - Reviewed supplementary information listed masks, glasses, gloves but these are broad and not specific kind to confirm relevance to the VOCs measured

Table 1 mentions other engineering of 32 % of the controls, which is the highest percentage of all. Can the author elaborate on these?

Line 564 The conclusion should be revised to account for the study limitations, including that future interventions to include quantification of TVOCs using reliable methodology and equipment as well as a representative sample size.

This work would have been more suitable to be summarised as a short communication than a full article due to the limitations of the study.

Reviewer #2: Review Questions

1. Is the manuscript technically sound, and do the data support the conclusions?

Yes, the conclusions are technical sound, and the data support the conclusions

Reasons:

Cluster randomized controlled trial design is suitable for evaluating Community health worker intervention to reduce worker exposure to volatile organic compounds in small business auto and beauty shops in a marginalized community.

The conclusions of the study are supported by the analysis and the data presented.

2. Has the statistical analysis been performed appropriately and rigorously?

Yes, the statistical analysis of the study is appropriate, well justified, and implemented.

Authors has used linear mixed-effects models to accounts for clustering at the shop level and the use of repeated measurements.

3. Have the authors made all data underlying the findings in their manuscript fully available?

Yes, but with justified restrictions.

The authors gave a clear and detailed Data Availability Statement. The restriction is for public sharing due to participant privacy. Since the data are not publicly available or deposited in an open repository, this may limit reproducibility of the study.

Is the manuscript presented in an intelligible fashion and written in standard English?

Yes, the manuscript is clear, well-presented and structured, and written in an acceptable scientific English.

Areas for improvement to the authors

The manuscript is lengthy and dense in some sections (particularly Methods and Results), which may challenge non-specialist readers.

The are occasional typographical errors that includes formatting inconsistencies (e.g., spacing, spelling, extra or missing spaces line breaks) should be corrected.

6. PLOS authors have the option to publish the peer review history of their article (what does this mean?). If published, this will include your full peer review and any attached files.

Reviewer #1: No

Reviewer #2: No

---

## [Author Response · Author response to Decision Letter 1]

9 Mar 2026

Response to Reviewers

Journal Requirements:

Response: We updated the manuscript to comply with PLOS ONE’s style requirements by doing the following:

• Changed all major section headings to Level 1 headings with bold 18-pt font in sentence case along with appropriate changes for Level 2 and 3 headings

• Double-spaced text

• Renamed figure files to Fig1.tif, etc.

• Renamed supporting information files and moved “Supporting information” section after “References” section

• Changed author affiliations from superscript letters to superscript numbers and did not use abbreviations

• Moved table legends to be above table footnotes

• Removed “Competing interests” and “Funding” subsections from manuscript (provided in online submission form)

• Moved “Ethics approval” and “Declaration of use of generative AI and AI-assisted technologies” subsections to subsections in the “Methods” section

[This project was supported by the National Institute of Environmental Health Sciences grants R01 ES028250, P30 ES006694, T32 ES007091, and R25 ES025494. The publication’s contents are solely the authors’ responsibility and do not necessarily represent the official views of the National Institutes of Health.].

Response: Our amended Funding Statement has been included in the revised cover letter.

3. Thank you for stating the following in your manuscript:

[This project was supported by the National Institute of Environmental Health Sciences grants R01 ES028250, P30 ES006694, T32 ES007091, and R25 ES025494. The publication’s contents are solely the authors’ responsibility and do not necessarily represent the official views of the National Institutes of Health.]

[This project was supported by the National Institute of Environmental Health Sciences grants R01 ES028250, P30 ES006694, T32 ES007091, and R25 ES025494. The publication’s contents are solely the authors’ responsibility and do not necessarily represent the official views of the National Institutes of Health.]

Response: We removed funding-related text from the manuscript, and our amended Funding Statement has been included in the cover letter. It is as follows: “This project was supported by the National Institute of Environmental Health Sciences grants R01 ES028250, P30 ES006694, T32 ES007091, and R25 ES025494. The publication’s contents are solely the authors’ responsibility and do not necessarily represent the official views of the National Institutes of Health. There was no additional external funding received for this study.”

4. In the online submission form you indicate that your data is not available for proprietary reasons and have provided a contact point for accessing this data. Please note that your current contact point is a co-author on this manuscript. According to our Data Policy, the contact point must not be an author on the manuscript and must be an institutional contact, ideally not an individual. Please revise your data statement to a non-author institutional point of contact, such as a data access or ethics committee, and send this to us via return email. Please also include contact information for the third-party organization, and please include the full citation of where the data can be found.

Response: Our “Availability of data and materials” statement has been revised to the following to include a non-author institutional point of contact: “Individual participant data cannot be made publicly available to protect participant privacy. The informed consent process did not include permission for public data sharing, and the dataset contains sensitive workplace and demographic information that could potentially identify participants in small businesses. Data access can be requested using the University of Arizona's Data Use Agreement (DUA) procedures from the Office of Research and Partnerships: https://research.arizona.edu/faq-page/data-use-agreement. Contact email to request a DUA: contracting@email.arizona.edu.”

Response: Our ethics statement has been moved to the Methods section: “Informed consent was obtained from all participants, and study ethics approval was obtained from the University of Arizona’s Human Subjects Protection Program (#1709821542), in accordance with the Declaration of Helsinki.”

6. We note that you have uploaded Supporting Information figures/tables that were not cited in your manuscript. Please update any in-text citations for your Supporting Information files in your manuscript. Please see our Supporting Information guidelines for more information: http://journals.plos.org/plosone/s/supporting-information.

Response: Citations for “S1 Protocol” and “S2 Checklist” have been added in the manuscript text.

Response: N/A

Response: We have reviewed the reference list to ensure it is complete and correct. We have not cited retracted papers, and we have not changed the references.

Reviewers' comments:

Reviewer's Responses to Questions

Comments to the Author

1. Is the manuscript technically sound, and do the data support the conclusions?

Reviewer #1: Partly

Reviewer #2: Yes

2. Has the statistical analysis been performed appropriately and rigorously?

Reviewer #1: Yes

Reviewer #2: Yes

3. Have the authors made all data underlying the findings in their manuscript fully available?

Reviewer #1: Yes

Reviewer #2: Yes

4. Is the manuscript presented in an intelligible fashion and written in standard English?

Reviewer #1: Yes

Reviewer #2: Yes

5. Review Comments to the Author

Reviewer #1: Firstly, this was a great study to support the marginalized workers with approaches to providing occupational health guidance

However, I have the following questions or request for clarity on the following:

Overall - the study has many limitations regarding methodology and sample size subjecting it not to make strong conclusion about the outcome. The equipment used in this study also had few limitations including placing them at the measurement sites ensuring proper sampling technique.

Response: We appreciate the reviewer's careful consideration of study limitations. We have discussed methodological constraints (including PID placement and sampling technique) and sample size in the Discussion/Limitations subsection. We believe our conclusions are appropriately tempered given these limitations. We have further revised the final sentence of the Conclusions (see response to comment on Conclusions below) to explicitly acknowledge the need for enhanced methodology and larger sample sizes in future research.

The study compared auto repair shops and beauty salons. However, it is not mentioning about reviewing the GHS safety data sheets of chemicals used at these two different facilities. Instead, the study checked if the chemical were unlabelled or not, but no information contained in the label was reviewed.

Response: We have clarified in the Methods/Intervention section that product labels and GHS safety data sheets were systematically reviewed. The original text stated: "For the intervention, CHWs assessed shop services, reviewed product inventories for VOC content, and suggested VOC exposure controls."

We have revised this to: "Before the intervention, the UA measurement team prepared product inventories by reviewing product labels, GHS safety data sheets (and for beauty salons, the EWG Skin Deep® Cosmetics Database), and entering these data into a SERI database for CHWs to use. For the intervention, CHWs assessed shop services, reviewed product inventories for VOC content, and suggested VOC exposure controls."

Not clear if participants were trained to look after the samplers before the study could take off since the researchers only visited three times to check and assist with questionnaires. It was not confirmed if the CHW had training to perform these measurements, equipment operation and they would be in a position to apply professional judgement similar to if it was done by the Industrial Hygienist for proposing industrial l hygiene interventions.

Response: The Methods/Trial design subsection originally stated: "For each workshift, the UA assessment team would usually visit the shop at the start of the workshift to set up, once in the middle to check on the monitors and help participants complete missing information on the log within that time frame, and then at the end of the workshift for final data and instrument collection."

It has been revised to the following to clarify how the UA assessment team trained participants to use the total VOC monitors: “For each workshift, the UA assessment team would visit the shop at the start of the workshift to turn on and set up the monitors for logging. At this initial visit, the team provided participants with a brief tutorial on monitor operation, demonstrated troubleshooting procedures (e.g., how to restart the monitor by holding the center button when it alarmed with a flashing red light), and left contact information. The team would visit once in the middle of the workshift to check on the monitors and help participants complete missing information on the log, and then at the end of the workshift for final data and instrument collection. Participants were instructed to contact the UA team if the monitors turned off, stopped making their typical sampling noise (low hum), or there were other issues they could not resolve.”

Additionally, we clarify that the trained researchers on the UA assessment team (not CHWs) were responsible for all exposure monitoring and equipment operation. CHWs provided industrial hygiene guidance and helped shops select exposure controls, but they did not operate monitoring equipment. CHWs received training in industrial hygiene principles, workplace assessment, and culturally appropriate engagement strategies relevant to their role in the intervention.

Line 320 referred to Supplementary material on PPE - Reviewed supplementary information listed masks, glasses, gloves but these are broad and not specific kind to confirm relevance to the VOCs measured

Response: We have added clarification in the Results/Intervention section after the discussion of PPE choices: “Although CHWs suggested PPE last according to the hierarchy of controls, the community-based approach required honoring owner requests for PPE appropriate to the specific tasks and products, which addressed occupational safety needs extending beyond VOC exposure alone.” As discussed in the Discussion section, these different control choices between sectors may partially explain the differential intervention effects on airborne VOC concentrations.

Table 1 mentions other engineering of 32 % of the controls, which is the highest percentage of all. Can the author elaborate on these?

Response: We thank the reviewer for pointing out that since there is such a large fraction in the local engineering controls “Other” category, it should be clarified. We have revised Table 1 to provide more detailed information about this category. We reclassified the free-text “Other” responses as follows: (1) all fan types (ceiling, box, desk, utility, portable, standing, and floor fans) were combined into a single “Fan” category, as these different fan types serve the same ventilation function and the original granular categorization was inconsistent (pre-defined fan types had individual checkboxes while additional fan types mentioned in free text were lumped into “Other”); (2) air purifiers were extracted into a new “Air purifier” category; and (3) items representing general ventilation rather than local engineering controls (open doors/windows, working outside, and coolers) were removed from this section to avoid duplication, as they are already captured in the “Types of ventilation” section of Table 1. This recategorization completely eliminated the “Other” category. A footnote has been added: “Fans includes any of ceiling fan, box fan, desk fan, utility fan, portable fan, standing fan, or floor fan.” W

---

## [Editor Report · Decision Letter 1]

18 Mar 2026

Community health worker intervention to reduce worker exposure to volatile organic compounds in small business auto and beauty shops in a marginalized community: a cluster randomized controlled trial

PONE-D-25-61403R1

Dear Dr. Gutenkunst,

We’re pleased to inform you that your manuscript has been judged scientifically suitable for publication and will be formally accepted for publication once it meets all outstanding technical requirements.

Kind regards,

Rajeev Singh

Academic Editor

PLOS One
---

## [Editor Report · Acceptance letter]

PONE-D-25-61403R1

PLOS One

Dear Dr. Gutenkunst,

I'm pleased to inform you that your manuscript has been deemed suitable for publication in PLOS One. Congratulations! Your manuscript is now being handed over to our production team.

Kind regards,

on behalf of

Dr. Rajeev Singh

Academic Editor

PLOS One